# Fully Neural Network Based Speech Recognition on Mobile and Embedded Devices

**Jinhwan Park**
Seoul National University
bnoo@snu.ac.kr

**Yoonho Boo**
Seoul National University
dnsgh@snu.ac.kr

**Iksoo Choi**
Seoul National University
akacis@snu.ac.kr

**Sungho Shin**
Seoul National University
ssh9919@snu.ac.kr

**Wonyong Sung**
Seoul National University
wysung@snu.ac.kr

## Abstract

Real-time automatic speech recognition (ASR) on mobile and embedded devices has been of great interests for many years. We present real-time speech recognition on smartphones or embedded systems by employing recurrent neural network (RNN) based acoustic models, RNN based language models, and beam-search decoding. The acoustic model is end-to-end trained with connectionist temporal classification (CTC) loss. The RNN implementation on embedded devices can suffer from excessive DRAM accesses because the parameter size of a neural network usually exceeds that of the cache memory and the parameters are used only once for each time step. To remedy this problem, we employ a multi-time step parallelization approach that computes multiple output samples at a time with the parameters fetched from the DRAM. Since the number of DRAM accesses can be reduced in proportion to the number of parallelization steps, we can achieve a high processing speed. However, conventional RNNs, such as long short-term memory (LSTM) or gated recurrent unit (GRU), do not permit multi-time step parallelization. We construct an acoustic model by combining simple recurrent units (SRUs) and depth-wise 1-dimensional convolution layers for multi-time step parallelization. Both the character and word piece models are developed for acoustic modeling, and the corresponding RNN based language models are used for beam search decoding. We achieve a competitive WER for WSJ corpus using the entire model size of around 15MB and achieve real-time speed using only a single core ARM without GPU or special hardware.

## 1 Introduction

Recently, neural network technology has greatly improved the accuracy of automatic speech recognition (ASR), and many applications are being developed for smartphones and intelligent personal assistants. Many researches on end-to-end speech recognition are being conducted to replace the hidden Markov model (HMM) based technique which has been used for many years. The end-to-end models with neural networks include connectionist temporal classification (CTC)-trained recurrent neural networks (RNN) [1, 2], encoder-decoder architectures [3, 4], and RNN transducers [5, 6]. Although HMM-based algorithms can be considered arithmetic efficient, they demand many irregular memory accesses and require a large memory foot-print usually exceeding a few hundred MBs. In contrast, RNN based ASR has the advantage of low memory footprint; however, it demands many arithmetic operations for real-time inference. Consequently, server-based ASR implementations are mostly used for real services, which however has the problem of response delay and user privacy is-

sues. Therefore, there is a huge demand for on-device ASR implementation not only for smartphones but also for many internet of things (IoT) devices [7].

There have been many researches on accelerating the inference of neural networks by employing special-purpose hardware or GPUs. Our estimate of a fully neural network based single user speech recognition demands about 1 Giga arithmetic operations per second, which is not a formidable barrier because a single instruction multiple data (SIMD) instruction can conduct four to eight arithmetic operations at a time and the clock frequency of a CPU is around 1 GHz. The real problem is the cache misses because the model size of RNN, which is over 10 MB for most ASR, is usually much larger than the cache size of embedded CPUs. For example, ARM Cortex-A57 has a 2 MB L2 cache at most. Due to the sequential nature of RNN, the parameters should be fetched from the DRAM at each time step, which implies continuous cache misses. In GPU or server-based implementations, this problem is hidden by batch or multi-stream parallel processing.

In embedded devices, however, only one stream is executed, because it usually targets a single user. To solve the memory access problem, we apply multi-time step parallel processing for which multiple consecutive frames are computed concurrently, and the number of DRAM accesses can be reduced in proportion to the number of parallelization steps. Unfortunately, the multi-time step parallelization cannot be applied to popular RNN structures, such as long short-term memory (LSTM) [8] or gated recurrent unit (GRU) [9], because they contain complex input-output dependencies. Recent studies demonstrated linear RNNs with simplified feedback, which can be used for multi-time step parallelization [10–13]. Quasi RNN (QRNN) and simple recurrent unit (SRU) are a kind of linear RNNs. However, when applied to acoustic modeling, the accuracy with linear RNN was not as good as that of LSTM RNN. We combined depth-wise 1-dimensional (1-D) convolution with linear RNN and obtained very good accuracy exceeding that of LSTM with a comparable parameter size.

The developed speech recognition system employs RNN based language models (LMs) instead of n-gram based ones. Character and word piece based LMs are used for beam search decoding. In addition, we also try hierarchical character LM (HCLM) for improved performance, and by which we can achieve an accuracy comparable to Deep Speech 2 [2] for Wall Street Journal (WSJ) corpus, with 10 times less parameters. The RNN LMs are based on LSTM or GRU because multiple streams are executed concurrently for beam search decoding. We reduce the overhead of DRAM accesses by executing multiple streams of RNN LMs at a time, where the stream size depends on the beam search width. For efficient beam search decoding, we early prune the output symbols of low probability in the acoustic model (AM).

The implementation operates in real-time on the ARM Cortex-A57 based embedded system without GPU support. The model size of the proposed speech recognition system including CTC-AM and RNN LM is about 15 MB with 8-bit parameters which is far smaller than that of conventional HMM-based systems.

This paper is organized as follows. In Section 2, we review the related works and recently proposed RNNs. We introduce the proposed speech recognition system in Section 3. The experimental results including the execution time analysis are shown in Section 4. Section 5 concludes this paper.

## 2   Related Works

Most mobile speech recognition methods have relied on WFST (weighted finite state transducer) based algorithms mainly because of their low arithmetic requirements [14]. However, a WFST network usually demands a foot-print of more than a few hundred MB because of the integrated n-gram based LM. Scattered and unaligned memory accesses also hinder efficient implementation of WFST networks.

Recently, fully neural network based speech recognition, which combines RNN based AM and LM, has drawn considerable attention. For efficient implementation of RNN, several model compression techniques have been developed, such as pruning [15], quantization [16], and matrix decomposition [17]. However, these techniques need efficient data rearrangement and decompression; therefore, they are not explored in this work. However, they can be combined with the proposed method. We applied 8-bit quantization to further reduce the execution time and the model size.

Recently, quasi RNN (QRNN) and simple recurrent unit (SRU) were developed for the purpose of fast training and inference of very long sequences [10, 11]. These RNNs only employ simple

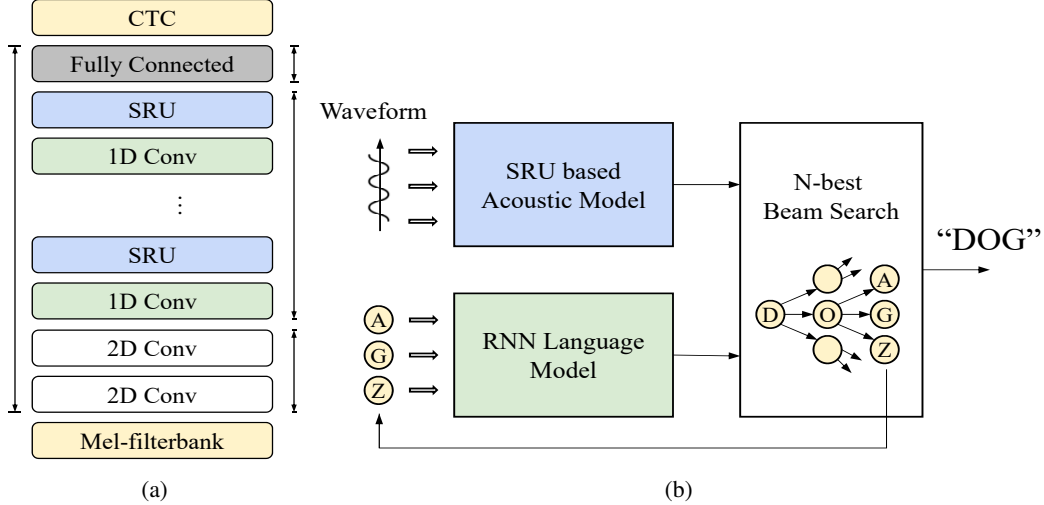

Figure 1: (a) The architecture of the neural network model used for acoustic modeling. (b) The system consists of RNN AM, RNN LM, and beam search decoding.

feedback that can be represented as linear recurrence equations, and allow not only fast training but also multi-time step parallel inference. QRNN showed a good performance comparable to LSTM in language modeling and machine translation.

Convolutional neural networks (CNNs) have also been used for sequence learning, such as machine translation [18, 19], language modeling [20], and speech recognition [21]. CNN has the advantage of parallel processing because there is no dependency between the input and output. However, CNN usually requires a large amount of feature maps, while RNN only needs to store the current cell and output state vectors in each layer. In this sense, we consider that RNN is more advantageous compared to CNN for on-device inference if multi-time step parallelization can be applied for memory access reduction.

## 3   Speech Recognition Algorithm

The target speech recognition system in this work consists of CTC-trained AM, RNN LM, and beam search decoder. In particular, we have developed two models; one is based on character as the output units of AM and LM, and the other is based on word piece units [22]. The word piece model is advantageous for real-time execution because the AM can operate at a low frame rate. In this section, we describe each component of the system in detail.

### 3.1   Acoustic modeling

The proposed AM architecture is shown in Figure 1 (a); it is composed of two 2-D convolutional layers, six recurrent layers, and the final fully connected layer. The output labels are either graphemes (characters) or word pieces. The AM RNN is trained with CTC loss [23]. The input of the convolutional layer consists of three 2-D feature maps with the time and frequency axes, and each feature map is formed with the mel-filter bank output, its delta, or double-delta. We employed two 2-D convolutional layers with a filter size of 5, as proposed in [2, 24]. The 2-D convolutional layers not only improve the recognition performance but also down-sample the input frames by two. Thus, the down-sampling in the convolutional layers not only reduces the computational complexity of the recurrent layers in AM but also simplifies the decoding.

The developed model contains six recurrent layers, and each layer consists of 1-D convolution unit and SRU. The recurrent layers adopt SRU which only employs cell-state feedback [11]. Given an input vector $\mathbf{x}_t \in \mathbb{R}^{d_{in}}$, the output $\mathbf{h}_t \in \mathbb{R}^N$ and the cell state $\mathbf{c}_t \in \mathbb{R}^N$ is computed in SRU as depicted in Equation (1). Inspired from *ifo-pooling* of QRNN, we replaced $(1 - \mathbf{f}_t)$ to input gate $\mathbf{i}_t$ [10]. We refer this as i-SRU in this work. In our experiments, adding the input gate not only improves

the robustness in training of SRU but also leads to a lower error rate. We also explored the location of $\tanh$ in our preliminary experiments, and found that locating $\tanh$ as shown in Equation (2) yielded slightly better results.

SRU:

$$
\begin{aligned}
\hat{\mathbf{x}}_t &= \mathbf{W}_z \mathbf{x}_t + \mathbf{b}_z, \\
\mathbf{f}_t &= \sigma(\mathbf{W}_f \mathbf{x}_t + \mathbf{b}_f), \\
\mathbf{o}_t &= \sigma(\mathbf{W}_o \mathbf{x}_t + \mathbf{b}_o), \\
\mathbf{c}_t &= \mathbf{f}_t \odot \mathbf{c}_{t-1} + (1 - \mathbf{f}_t) \odot \hat{\mathbf{x}}_t, \\
\mathbf{h}_t &= \mathbf{o}_t \odot \tanh(\mathbf{c}_t) + (1 - \mathbf{o}_t) \odot \mathbf{x}_t
\end{aligned}
\tag{1}
$$

i-SRU:

$$
\begin{aligned}
\hat{\mathbf{x}}_t &= \tanh(\mathbf{W}_z \mathbf{x}_t + \mathbf{b}_z), \\
\mathbf{f}_t &= \sigma(\mathbf{W}_f \mathbf{x}_t + \mathbf{b}_f), \\
\mathbf{i}_t &= \sigma(\mathbf{W}_i \mathbf{x}_t + \mathbf{b}_i), \\
\mathbf{o}_t &= \sigma(\mathbf{W}_o \mathbf{x}_t + \mathbf{b}_o), \\
\mathbf{c}_t &= \mathbf{f}_t \odot \mathbf{c}_{t-1} + \mathbf{i}_t \odot \hat{\mathbf{x}}_t \\
\mathbf{h}_t &= \mathbf{o}_t \odot \mathbf{c}_t + (1 - \mathbf{o}_t) \odot \mathbf{x}_t
\end{aligned}
\tag{2}
$$

where $\mathbf{W}_z, \mathbf{W}_f, \mathbf{W}_i, \mathbf{W}_o \in \mathbb{R}^{N \times d_{in}}$ and $\mathbf{b}_z, \mathbf{b}_f, \mathbf{b}_i, \mathbf{b}_o \in \mathbb{R}^N$ are trainable parameters. When the i-SRU is trained alone, the resulting WER is significantly worse than that of LSTM. To overcome this problem, we add a depth-wise 1-D convolutional layer at the input of each recurrent layer. This needs $O(k \times d_{in})$ additional parameters when the filter width of $k$ is used for the convolution. The number of parameters used in 1-D convolutional layers is much smaller than that for recurrent layers, which is $O(N \times d_{in})$. This is similar to QRNN with a filter size of $k > 1$. However, QRNN needs $O(k \times N \times d_{in})$ parameters. We obtained significant performance improvement by adding 1-D convolutional layers between the recurrent layers. Specific results are reported in Section 4.

The multi-time step processing converts matrix-vector multiplication into matrix-matrix multiplications. The weight matrix is reused for $T$ time steps with a single parameter fetch from the DRAM, by which the execution time and power consumption can be greatly reduced. The multi-time step computation is shown in Equation (3), where $T$ is the number of parallelization steps.

$$
\begin{bmatrix}
\mathbf{z}_1 & \mathbf{z}_2 & \dots & \mathbf{z}_T \\
\mathbf{i}_1 & \mathbf{i}_2 & \dots & \mathbf{i}_T \\
\mathbf{f}_1 & \mathbf{f}_2 & \dots & \mathbf{f}_T \\
\mathbf{o}_1 & \mathbf{o}_2 & \dots & \mathbf{o}_T
\end{bmatrix}
=
\begin{pmatrix}
\mathbf{W}_z \\
\mathbf{W}_i \\
\mathbf{W}_f \\
\mathbf{W}_o
\end{pmatrix}
(\mathbf{x}_1 \ \mathbf{x}_2 \ \dots \ \mathbf{x}_T)
\tag{3}
$$

## 3.2 Character-based model

RNN based LMs show quite high performance when compared to statistical n-gram based LMs. When AM employs characters as the output unit, a dictionary or character-level LM (CLM) can be used for reducing the word error rate (WER) [25]. The CLM only supports 30 labels; therefore, the input and output layers are very simple. In addition, the CLM does not have the out-of-vocabulary (OOV) problem. We do not use the word-level LM (WLM) because it has the OOV problem and consumes a large number of parameters at the output softmax layer. Instead of adopting WLM, we use the hierarchical character-level LM (HCLM) for further improving the performance [26]. Note that the HCLM consists of two RNN modules: one operates with the character clock and the other with the word clock. Because the RNN modules operate with the word-clock, the HCLM can show very low bit-per-character (bpc) performance.

The LMs are used in beam search decoding, and the maximum number of LMs that operate simultaneously depends on the beam size, which is between 32 and 128 in this system. This suggests the use of multi-stream parallel processing to improve the execution speed of LMs. Note that multi-stream parallel processing executes multiple sequences concurrently, while multi-time step parallel processing, which is adopted in AM RNN, computes multiple output samples at a time. Thus, conventional RNN, such as LSTM or GRU models, can be used for LM design. We need to save the context of all the LMs during the beam search decoding. GRU has fewer states to keep than LSTM. Therefore GRU can reduce memory consumption in the decoding stage. See Appendix A for the block diagram of HCLM.

### 3.3 Word piece-based model

We have also developed a word piece model based ASR to reduce the complexity further by lowering the frame rate. The word piece model includes very frequently used words, sub-words, and characters [22]. The number of word pieces used in this system is 500 and 1,000. The word piece model does not have the OOV problem because any word can be constructed using sub-words or characters. The performance of the word piece LM can be better than that of CLM because a word piece is usually composed of several characters, which implies an ability to predict longer dependency than CLM. The word piece model is very advantageous for reducing the complexity because the AM with word piece can operate at a slower rate when compared to the AM with character. In addition to the frame rate down-sampling at the convolutional layer, we apply down-sampling in the recurrent layers also. However, word piece model training, especially for AM, demands much more data because there are an increased number of labels.

### 3.4 Decoding

For inference, we find the sequence of label $\mathbf{y}$, which maximizes $Q(\mathbf{y})$ for a given input feature $\mathbf{x}_{1:T}$, by combining the output of AM $P_{CTC}$ and LM $P_{LM}$ as follows

$$Q(\mathbf{y}) = \log(P_{\text{CTC}}(\mathbf{y}|\mathbf{x}_{1:T})) + \alpha \log(P_{\text{LM}}(\mathbf{y})) + \beta|\mathbf{y}| \tag{4}$$

The labels can be either characters or word pieces. We use the beam search decoding algorithm for incremental speech recognition as proposed in [25]. The computational complexity of beam search decoding with RNN LM is O(*beam width × transcription length × vocabulary size*). To decrease the search space of decoding, we applied two techniques. First, we skip the decoding process for the current input when the blank output probability is larger than 0.95. This removes unnecessary search caused by blank frames [27]. Second, we sort the AM output probability and conduct decoding for top-$k$ probability labels only. This is especially effective for word piece models because the vocabulary size of word pieces is at least 10 times larger than that of the character level LM. We used $k = 10$ for word piece model. Please refer Appendix C for the details of the decoding algorithm.

Table 1: WER and CER in percentage on WSJ eval92 test set. Decoding is conducted with RNN CLM and HCLM.

| | | Greedy | | CLM | | HCLM | |
|---|---|---|---|---|---|---|---|
| **Model** | **Params.** | **CER** | **WER** | **CER** | **WER** | **CER** | **WER** |
| 6x800 SRU | 10.62M | 26.94 | 82.56 | 13.24 | 29.68 | 7.94 | 15.41 |
| 6x700 i-SRU | 10.92M | 12.70 | 45.22 | 7.04 | 18.90 | 4.90 | 12.27 |
| 6x800 SRU, 1-D conv | 10.69M | 6.06 | 22.16 | 3.48 | 9.53 | **1.97** | **4.90** |
| 6x700 i-SRU, 1-D conv | 10.98M | **5.26** | **19.07** | **2.70** | **7.30** | 2.01 | **4.90** |
| 6x1000 i-SRU, proj, 1-D conv | 14.14M | 5.85 | 21.60 | 3.00 | 7.80 | 2.27 | 5.17 |
| 4x600 LSTM | 10.85M | 7.29 | 24.88 | 5.35 | 14.27 | 3.70 | 8.75 |
| 4x600 LSTM, 1-D conv | 10.88M | 6.95 | 23.57 | 5.80 | 15.22 | 3.10 | 7.01 |
| 4x840 LSTM, proj, 1-D conv | 12.01M | 7.78 | 26.80 | 4.88 | 12.26 | 3.36 | 7.60 |
| 6x300 Gated ConvNet | 16.38M | 8.02 | 28.65 | 5.13 | 13.82 | 2.98 | 6.74 |
| 4x550 GILR-LSTM | 11.34M | 8.60 | 31.99 | 4.86 | 13.60 | 2.66 | 6.35 |
| 4x550 GILR-LSTM, 1-D conv | 11.37M | 7.15 | 26.06 | 4.44 | 11.92 | 2.38 | 5.45 |
| *bidirectional models* | | | | | | | |
| 6x400 i-SRU, 1-D conv | 11.52M | **4.90** | **17.30** | **2.94** | **7.90** | **1.97** | **4.87** |
| 4x350 LSTM | 10.70M | 5.88 | 20.17 | 3.46 | 9.41 | 2.57 | 5.89 |

## 4  Experimental Results

We present the AM training results on character and word piece models. The decoding is conducted with RNN LMs. In addition, the execution time is analyzed.

Table 2: Comparision of the model with non-causal and causal 1-D convolutions. 1-D conv (-$a$, $b$) uses $a$ past and $b$ future time-steps to compute the output of the current time step.

| | | Greedy | | CLM | | HCLM | |
|---|---|---|---|---|---|---|---|
| **Model** | | **CER** | **WER** | **CER** | **WER** | **CER** | **WER** |
| 6x700 i-SRU, 1-D conv (-7, 7) | | 5.26 | 19.07 | 2.70 | 7.30 | 2.01 | 4.90 |
| 6x700 i-SRU, 1-D conv (-14, 0) | | 5.70 | 20.18 | 3.12 | 8.47 | 2.30 | 5.32 |
| 6x700 i-SRU, 1-D conv (-7, 0) | | 6.10 | 21.96 | 2.99 | 7.69 | 2.35 | 5.55 |

Table 3: WER and CER in percentage on WSJ eval92 test set when trained with additional data.

| | | Greedy | | CLM | | HCLM | |
|---|---|---|---|---|---|---|---|
| **Model** | **Params.** | **CER** | **WER** | **CER** | **WER** | **CER** | **WER** |
| 6x700 i-SRU, 1-D conv | 10.98M | 4.13 | 18.02 | 2.54 | **6.04** | 1.51 | 3.73 |
| 6x1000 i-SRU, proj, 1-D conv | 14.14M | **3.80** | 14.70 | **2.19** | 6.20 | **1.48** | **3.70** |
| 4x600 LSTM, 1-D conv | 10.88M | 4.35 | **13.90** | 3.72 | 10.15 | 2.55 | 5.92 |
| 4x840 LSTM, proj, 1-D conv | 12.01M | 5.76 | 20.15 | 3.54 | 9.25 | 2.53 | 5.79 |
| Deep Speech 2 | 100M | WER 3.60 with 5-gram LM | | | | | |

## 4.1 Acoustic models

The most critical part of this research is the development of an AM using linear RNN, such as SRU or QRNN. In our early experiments, we failed to obtain good performance by only using SRU or QRNN for recurrent layers. Therefore, we needed to try many different RNN structures, thus each training should not take much time. We constrained the number of parameters to be approximately 12M. The models used for performance evaluation include the conventional LSTM, SRU, Gated ConvNet, and GILR-LSTM [12]. In addition, we trained each model with the 1-D convolution at the input. The width of 1-D convolution is set to 15, which seems to be the optimum number at our experiments. Note that the 1-D convolution considers 7 past (-7) and 7 future (+7) time-steps unless specified otherwise. We used uni-directional models because this implementation is intended for online speech recognition. Bi-directional models for i-SRU and LSTM are also included for performance comparison.

We used Wall Street Journal (WSJ) SI-284 training set (81 hours) for the fast evaluation of AMs. A 40-dimensional log mel-frequency filterbank feature was extracted from the raw speech data. The feature vectors were sampled every 10 ms with 25 ms Hamming window. We applied batch normalization [28] to the first two convolutional layers and variational dropout [29] to every output of the recurrent layer for regularization. Adam optimizer [30] was applied for training. We used an initial learning rate of 3e-4, and the learning rate was reduced to half if the validation error was not lowered for consecutive 8 epochs. Gradient clipping with a maximum norm of 4.0 was applied. For comparison, we trained all the models with identical hyper-parameter setting.

The trained models were evaluated on WSJ eval92 set. The beam search decoding was conducted using the same CLM or the same HCLM. The CLM used for decoding consists of two-layer 512-dimensional LSTM. The HCLM has four recurrent layers, where two layers are assigned to the word-level modeling [26]. RNN LM was trained on WSJ LM training text. We randomly selected 5% of WSJ LM training text to the valid set, and another 5% to the test set. The remaining 90% of the text is used for training RNN LM. The RNN LM reported 1.20 of bit-per-character (bpc) on the test set, while the HCLM showed a bpc of 1.07 on the test set. The decoding was conducted with a beam width of 128.

Table 1 shows the CER and WER performance of the RNN models trained with the WSJ SI-284 training set. To denote the projection, 'proj' is used for each RNN model. The size of the projection-layer is a half of the RNN dimension. For example, the LSTM with the layer size of 840 employs the projection dimension of 420. The table includes the results of greedy, CLM, and HCLM based

decoding. Here, we can find that i-SRU with 1-D convolution performs much better than LSTM. The performance of SRU with 1-D convolution is not much different from that of i-SRU with 1-D convolution. In this table, we can also find that HCLM helps considerably in reducing the CER and WER for all models.

Since the 1-D convolution layers consider the future inputs for improved performance, the output is not generated immediately. We trained i-SRU with causal depth-wise 1-D convolutions and the results are shown in Table 2. The performances of some other models that employ different observation windows are also shown. Note that the WER of the model with the causal 1-D convolution is still much lower than that of LSTM.

Since the amount of data is critical, we further trained two selected models, i-SRU and LSTM, using all the available speaker independent data in the WSJ corpus to improve the WER. This corresponds to approximately 167 hours of speech. The results are shown in Table 3. WER of DeepSpeech2 is included for comparison, which is trained with more than 10,000 hours of data and the decoding is performed using 5-gram LM.

Table 4: WER and CER on WSJ eval92 when word piece units are used.

| | Greedy | | WPLM | |
| --- | --- | --- | --- | --- |
| **Model** | **CER** | **WER** | **CER** | **WER** |
| 6x700 i-SRU, 1-D conv | 7.37 | 17.95 | 6.73 | 10.50 |
| 4x600 LSTM, 1-D conv | 9.34 | 22.56 | 8.47 | 15.64 |
| 6x700 i-SRU, 1-D conv, additional data | 5.47 | 14.38 | 3.11 | 8.28 |
| 4x600 LSTM, 1-D conv, additional data | 6.57 | 15.32 | 4.53 | 11.48 |

Table 5: Comparison of WER and CER on WSJ eval 92 according to downsampling in the word piece AMs.

| | Greedy | | WPLM | |
| --- | --- | --- | --- | --- |
| **Model** | **CER** | **WER** | **CER** | **WER** |
| x2 in conv. layer | 7.02 | 18.95 | 6.05 | 10.93 |
| x4 in conv. layer | 8.05 | 20.24 | 6.55 | 11.83 |
| x2 in conv. layer, x2 in recurrent layer | 7.37 | 17.95 | 6.00 | 10.50 |
| x4 in conv. layer, x2 in recurrent layer | 10.30 | 25.58 | 7.83 | 13.99 |

## 4.2  Word piece based speech recognition

We trained the i-SRU with 1-D convolutional layer on the word piece model with a vocabulary size of 500. We also trained the LSTM model with the same setting to compare the CER and WER. We added the time-convolution with a stride of two just before the last two recurrent layers. This incurs x4 down-sampling in total. Decoding was conducted under a beam width of 64 with the word piece-level LM. The word piece LM consists of two layers of 512-dimensional LSTM, which has the same structure as that of RNN LM in Section 4.1.

The CER and WER for the word piece models are listed in Table 4. We noticed that the word piece models show higher WER when compared to the character-based models. One possibility is the data sparsity problem due to the small amount of training data in WSJ SI-284. To relieve this problem, we trained the word piece models using all speaker independent data in WSJ training set, which has 167 hours of data. We could obtain fairly reduced WER as shown in the table.

The difference between WER and CER in the word piece models is much smaller when compared to that in character based models. This suggests that WER improvement by beam search decoding is less significant in the word piece models. This leads to a smaller beam width in the decoding process, which also reduces the decoding complexity.

Table 6: WER and CER on Librispeech *test-clean* . The models are trained on LibriSpeech *train-clean-100* and *train-clean-360*.

| Model | Params. | Greedy CER | Greedy WER | RNN LM CER | RNN LM WER |
|---|---|---|---|---|---|
| 4x600 LSTM, character | 10.85M | 8.49 | 26.10 | 7.34 | 21.80 |
| 6x700 i-SRU, 1-D conv, character | 10.98M | 6.21 | 20.41 | 5.66 | 13.78 |
| 6x700 i-SRU, 1-D conv, word piece-500 | 11.30M | 6.72 | 17.10 | 4.67 | 9.98 |
| 6x700 i-SRU, 1-D conv, word piece-1000 | 11.65M | 6.62 | 16.16 | 4.42 | 9.61 |

Table 7: WER on Librispeech *test-clean* and *test-other*. The models are trained on all the LibriSpeech train set (960 hours).

| Model | Params. | test-clean | test-other | LM type |
|---|---|---|---|---|
| 6x700 i-SRU, 1-D conv | 12M | 9.02 | 23.60 | RNN LM |
| 12x1000 i-SRU, 1-D conv | 36M | 5.73 | 15.96 | RNN LM |
| Gated ConvNet [21] | 208M | 4.8 | 14.5 | 4-gram LM |
| 5-conv + 4x1024 bidirectional GRU [31] | 75M | 5.4 | 14.7 | 4-gram LM |
| Encoder-decoder [32] | 150M | 3.82 | 12.76 | RNN LM |

One advantage of employing the word piece unit is the possibility of more aggressive down-sampling. We analyzed the effect of down-sampling on WER in Table 5. The additional down-sampling is located in the second 2-D convolutional layer or before the last two recurrent layers. The lowest WER is reported when a down-sampling of 2 is applied to both the 2-D convolutional layer and the recurrent layer. Increasing the down-sampling ratio is very beneficial in reducing the decoding complexity.

We also trained our system using a larger dataset, Librispeech Corpus [33]. The AMs were trained with *train-clean-100* and *train-clean-360*. The training hyperparameters were exactly the same with the setting in Section 4.1. We trained both the character and word piece models. The word piece vocabulary sizes were 500 and 1,000, respectively. Two-layer 600-dimensional GRU was employed for RNN LM. The results of WERs on *test-clean* are shown in Table 6. The word piece model shows better performance than the character-level model unlike the WSJ dataset. This shows that the word piece model could be better if larger training dataset is available.

We further trained a large model with all the LibriSpeech training data (960 hours), which includes *train-other-500*. For the large model, we added 1-D convolutions at the input of every two SRU layers. The projection layers are used in each output of SRU layer. The word piece model with a vocabulary size of 1,000 is used. The WERs of the model are reported in Table 7. Ours achieves competitive WER compared to other models in recent works. Note that our model is unidirectional and has a far less number of parameters than other ones.

Table 8: Execution time of SRU-AM for 1 second of speech according to the number of parallelization steps.

| Parallelization Step | 1 | 2 | 4 | 8 | 16 | 32 |
|---|---|---|---|---|---|---|
| Computation time | 1.2129 | 0.6098 | 0.3065 | 0.2064 | 0.1524 | 0.1174 |

## 4.3 Execution time analysis

We present the implementation results of the proposed speech recognition models on the ARM Cortex-A57 based embedded system. The ARM CPU has 80 KB L1 data cache and 2,048 KB L2 cache. OpenBLAS library [34] is used for the optimization of computation. Table 8 shows the

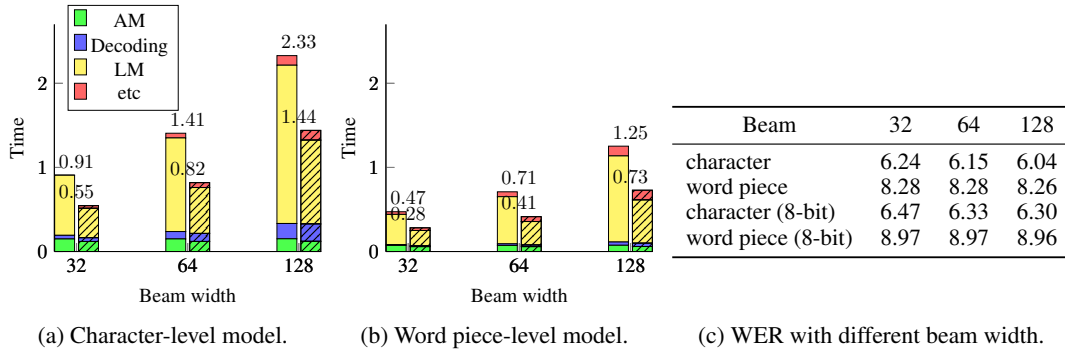

| (a) Character-level model. | (b) Word piece-level model. | (c) WER with different beam width. |

Figure 2: (a, b): Processing time of the speech recognition system for 1 second of speech on the single core ARM CPU. The time is evaluated on the WSJ eval92 dataset. The plot with dashed lines represents the computation time with 8-bit weights. (c): WERs when different beam width is used.

execution time of 6x700 i-SRU with 1-D convolution in Table 1 according to the number of multi-time steps for parallelization. When the number of parallel steps is 8, the execution time of AM for one second of speech is reduced to 0.2 sec, showing a speed-up of 6 times when compared to the single step execution.

Figure 2 shows the execution time estimate of the character and word piece models when the beam widths are 32, 64, and 128. The 6x700 i-SRU with 1-D convolution is used for AM of the system. Decoding is conducted with CLM or WPLM explained in Section 3.2. We also present the execution time when 8-bit weights are used for computation. For 8-bit implementation, gemmlowp library [35] is employed. The additional down-sampling allows the word piece model to run much faster than the character based model. Furthermore, the word piece model demands a much smaller beam width than the character model. By decreasing the beam width from 128 to 32, the computation time can be reduced to less than half while the difference in WER is only about 0.02%. Therefore, the word piece model is more advantageous for real-time speech recognition.

## 5 Concluding Remarks

Real-time automatic speech recognition (ASR) on embedded CPUs is studied by integrating end-to-end trained acoustic RNN, character or word piece language model RNN, and efficient decoding algorithm. To reduce the DRAM access overhead, we apply multi-frame parallel processing for the AM RNN, and develop high accuracy CTC-trained AM using simple recurrent units (SRUs) combined with 1-dimensional convolution at the input. We develop two ASR models; one employs character-based AM operating at 20 msec frame rate, and the other uses the word piece based AM that operates at the frame rate of 40 msec. The character based model shows very high accuracy on WSJ corpus when combined with the hierarchical character language model. The word piece based model shows x2 of the real-time speed on an ARM CPU mainly due to x4 down-sampling of the word piece AM. This study can be applied to all single stream or small batch-size implementation of ASR regardless of the platform, such as GPU or special-purpose hardware.

### Acknowledgments

This work was supported in part by the Brain Korea 21 Plus Project and the National Research Foundation of Korea (NRF) grant funded by the Korea government (MSIP) (No.2018R1A2A1A05079504).

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
