[Supplementary Material]

# A  Block diagram of hierarchical character-level language model

Figure 1: The architecture of the hierarchical recurrent neural network model used for language modeling.

# B  Formulation of recurrent units

The recurrent units used in this work are described as follows:

LSTM:
$$
\begin{aligned}
\mathbf{f}_t &= \sigma\left(\mathbf{W}_f \mathbf{x}_t + \mathbf{U}_f \mathbf{h}_{t-1} + \mathbf{b}_f\right), \\
\mathbf{i}_t &= \sigma\left(\mathbf{W}_i \mathbf{x}_t + \mathbf{U}_i \mathbf{h}_{t-1} + \mathbf{b}_i\right), \\
\mathbf{o}_t &= \sigma\left(\mathbf{W}_o \mathbf{x}_t + \mathbf{U}_o \mathbf{h}_{t-1} + \mathbf{b}_o\right), \\
\hat{\mathbf{c}}_t &= \tanh\left(\mathbf{W}_c \mathbf{x}_t + \mathbf{U}_c \mathbf{h}_{t-1} + \mathbf{b}_c\right), \\
\mathbf{c}_t &= \mathbf{f}_t \odot \mathbf{c}_{t-1} + \mathbf{i}_t \odot \hat{\mathbf{c}}_t, \\
\mathbf{h}_t &= \mathbf{o}_t \odot \tanh\left(\mathbf{c}_t\right).
\end{aligned}
\tag{1}
$$

GRU:
$$
\begin{aligned}
\mathbf{z}_t &= \sigma\left(\mathbf{W}_z \mathbf{x}_t + \mathbf{U}_z \mathbf{h}_{t-1} + \mathbf{b}_z\right), \\
\mathbf{r}_t &= \sigma\left(\mathbf{W}_r \mathbf{x}_t + \mathbf{U}_r \mathbf{h}_{t-1} + \mathbf{b}_r\right), \\
\mathbf{h}_t &= (1 - \mathbf{z}_t) \odot \mathbf{h}_{t-1} + \mathbf{z}_t \odot \tanh\left(\mathbf{W}_h \mathbf{x}_t + \mathbf{U}_h(\mathbf{r}_t \odot \mathbf{h}_{t-1}) + \mathbf{b}_h\right).
\end{aligned}
\tag{2}
$$

GILR-LSTM:
$$
\begin{aligned}
\mathbf{g}_t &= \sigma\left(\mathbf{V}_g \mathbf{x}_t + \mathbf{b}_g\right), \\
\mathbf{j}_t &= \tanh\left(\mathbf{V}_j \mathbf{x}_t + \mathbf{b}_j\right), \\
\tilde{\mathbf{h}}_t &= \mathbf{g}_t \odot \tilde{\mathbf{h}}_{t-1} + (1 - \mathbf{g}_t) \odot \mathbf{j}_t, \\
[\mathbf{f}_t, \mathbf{i}_t, \mathbf{o}_t] &= \sigma\left([\mathbf{U}_f, \mathbf{U}_i, \mathbf{U}_o]\tilde{\mathbf{h}}_{t-1} + [\mathbf{V}_f, \mathbf{V}_i, \mathbf{V}_o]\mathbf{x}_t + [\mathbf{b}_f, \mathbf{b}_i, \mathbf{b}_o]\right), \\
\mathbf{z}_t &= \tanh\left(\mathbf{U}_z \tilde{\mathbf{h}}_{t-1} + \mathbf{V}_z \mathbf{x}_t + \mathbf{b}_z\right), \\
\mathbf{c}_t &= \mathbf{f}_t \odot \mathbf{c}_{t-1} + \mathbf{i}_t \odot \mathbf{z_t}, \\
\mathbf{h}_t &= \mathbf{o}_t \odot \mathbf{c}_t.
\end{aligned}
\tag{3}
$$

## C  Decoding algorithm

**Input:** AM output probability matrix $P_{\text{CTC}} \in \mathbb{R}^{|\Sigma| \times T}$, beam width $B$, numer of search candidates $k$, language model weight $\alpha$, insertion bonus $\gamma$, vocabulary $\Sigma$

```
1  A_prev = {φ}
2  P_CTC(blank|x_{1:0}) = 1
3  for t = 1 to T do
4      if P_CTC(blank|x_{1:t-1}) > 0.95 and P_CTC(blank|x_{1:t}) > 0.95 then
5          continue
6      end
7      A_next = {}
8      K = top-k labels in Σ according to value of P_CTC(c|x_{1:t})
9      for l in A_prev do
10         for c in K do
11             if c = blank then
12                 p_nb(l) = p_nb(l)P_CTC(blank|x_{1:t})
13                 p_b(l) = (p_nb(l) + p_b(l))P_CTC(blank|x_{1:t})
14             else
15                 l⁺ = concat(l, c)
16                 if c = l_end then
17                     p_nb(l⁺) = p_b(l)P_CTC(c|x_{1:t})γP_LM(c|l)^α
18                     p_nb(l) = p_b(l)P_CTC(c|x_{1:t})
19                 else
20                     p_nb(l⁺) = (p_b(l) + p_nb(l))P_CTC(c|x_{1:t})γP_LM(c|l)^α
21                 end
22             end
23             add l⁺ to A_next
24         end
25     end
26     assign top-B of A_next to A_prev
27 end
```

**Algorithm 1:** Prefix beam search in proposed system.

The most time-consuming part in the decoding is computing the probability of $P_{LM}(c|l)$, which is required in the line 17 and 20. Time complexity of the LM computation is $O(B \times T \times |\Sigma|)$, but the actual computation complexity can be reduced to $O(B \times |l| \times |\Sigma|)$ by reusing the result of LM for same inputs. Skipping consecutive blanks and candidate pruning are applied in line number 5 and 8, respectively. Table 1 shows the ratio of skipped repeated blank frames. The number of operations of LM per frame according to the number of candidates are shown in Table 2.

Table 1: The ratio of frames whose decoding stages are skipped due to high CTC blank output.

| Acoustic model | Downsampling ratio | Percentage of skipping |
|---|---|---|
| WSJ - Character | $\times 2$ | 33.8% |
| WSJ - Word piece | $\times 4$ | 44.33% |
| Libri - Word piece | $\times 8$ | 20.23% |

Table 2: The number of LM operations with the varying number of candidates.

| Number of candidates | 20 | 30 | 40 | 100 |
|---|---|---|---|---|
| LM operations / frame | 4.365 | 4.488 | 4.593 | 4.820 |

# D    Training curves of the acoustic models

We compared the train and valid loss curves of some selected models. The peaks in training curves are due to curriculum-like learning scheduling. SRU without 1-D convolution was not trained well. LSTM was converged faster than other models but it reached local minimum quickly. Training loss of i-SRU was reduced faster than SRU, while they showed the similar valid loss in the end of training.

Figure 2: Training loss of acoustic models when trained on WSJ SI-284.

Figure 3: Validation loss of acoustic models when trained on WSJ SI-284

# E Example results of word piece and character level speech recognition

Table 3: Examples which are correct in wordpiece-level model but wrong in character-level model.

| Label | THE PINK SHEET A DRUG INDUSTRY TRADE LETTER REPORTED |
|---|---|
| char-greedy | THE PANKSHET A DRUG INDUSTRY TRADE LETTER REPORTED |
| char-LM | THE BANK HIT A DRUG INDUSTRY TRADE LETTER REPORTED |
| wp-greedy | THE PINK SHEET A DRUG INDUSTRY TRADE LETTER REPORTED |
| wp-LM | THE PINK SHEET A DRUG INDUSTRY TRADE LETTER REPORTED |
| Label | THIS WEEK LOCAL GOVERNMENTS HAVE APPEARED IN THE... |
| char-greedy | THIS WEEK LOCAL GOVERNMENTS HAVE THE PEAR IN THE... |
| char-LM | THIS WEEK LOCAL GOVERNMENTS HAVE THE PART IN THE... |
| wp-greedy | THIS WEEK LOCAL GOVERNMENTS HAVE APPEARED IN THE... |
| wp-LM | THIS WEEK LOCAL GOVERNMENTS HAVE APPEARED IN THE... |

Table 4: Examples that are correct in character-level model but wrong in word piece-level model.

| Label | SO MR. WANG TELLS PEOPLE HE IS FIFTY |
|---|---|
| char-greedy | SO MR. WEANGTELL'S PEOPLE HE IS FIFTY |
| char-LM | SO MR. WANG TELLS PEOPLE HE IS FIFTY |
| wp-greedy | SO MR. WGTELLES PEOPLE HE IS FIFTY |
| wp-LM | SO MR. WANGTELL'S PEOPLE HE IS FIFTY |
| Label | THE MARKETS TEND TO MAGNIFY THE NEWS |
| char-greedy | THE MARKETS TEND DOMAGNIFY THE NEWS |
| char-LM | THE MARKETS TEND TO MAGNIFY THE NEWS |
| wp-greedy | THE MARKET'S TEND TO MAGNIFY THE NEWS |
| wp-LM | THE MARKET'S TEND TO MAGNIFY THE NEW |

# F Example of results that are corrected when HCLM is used

Table 5: Example sentences that are corrected when HCLM is used.

| Label | RESPONSES TEND TO BE MUTE |
|---|---|
| greedy | RESPONSEES TEND TO BE MUNED |
| LSTM-CLM | RESPONSE IS TEND TO BE MUTED |
| HCLM | RESPONSES TEND TO BE MUTED |
| Label | I'M NOT AT ALL UNHAPPY WITH WHAT I'M SEEING |
| greedy | I'M NOT IT ALL AND HAPPY WITH WHAT I'M SEE |
| LSTM-CLM | I'M NOT AT ALL AND HAPPY WITH WHAT I'M SEEN |
| HCLM | I'M NOT AT ALL UNHAPPY WITH WHAT I'M SEEING |
| Label | THE BIG SHOE IS GOING TO DROP WHEN WE SEE THE TRADE NUMBER |
| greedy | THE BIG SHE WAS GOING TO DROP INLY SEE THE TRADE NUMBER |
| LSTM-CLM | THE BIG SHE WAS GOING TO DROP IN WE SEE THE TRADE NUMBER |
| HCLM | THE BIG SHOE IS GOING TO DROP WHEN WE SEE THE TRADE NUMBER |
| Label | AND THEY BALK AT THE APPROACH USED IN MEAT AND POULTRY PLANTS... |
| greedy | AND THEY BALCK AT THE APPROACH USED IN MEET AND PULTRYY PLANTS... |
| LSTM-CLM | AND THEY BALK AT THE APPROACH USED IN MEET AND POULTRY PLANTS... |
| HCLM | AND THEY BALK AT THE APPROACH USED IN MEAT AND POULTRY PLANTS... |