[Reviews · NeurIPS 2018]

Reviewer 1



This is a good paper that proposes acoustic modeling techniques for speech recognition on mobile and embedded devices. I can see the following highlights: 1. The authors propose a multi-time step parallelization approach that computes multiple output samples at a time with the parameters fetched from the DRAM to reduce the number of accesses to DRAM. 2. The authors achieve good speech recognition performance on WSJ and Librispeech datasets using models running on a single core of Cortex A-57. 3. The authors provide relative complete experiment results including CER, WER, real time factor, breakdowns of computation cost, and so on. But I also have the following suggestions: 1. From what I can see in the paper, after optimizing the acoustic model, the computation of the LM part now dominates the total computation time. To make this really useful for mobile or embedded devices, the authors will have to tackle the LM issue as well. 2. The performance on Librispeech is good, but not excellent. And I would also be interested in seeing results on the noisy datasets from Librispeech. 3. It will be great if the authors can cite state-of-the-art results on WSJ and Librispeech just for the purpose of comparison. Overall, I think this is a good paper that can potentially guide the direction of speech recognition on mobile and embedded devices.

Reviewer 2



The authors propose a novel RNN ASR system that combines previously proposed simple recurrent units (SRU) and ifo-pooling. By incorporating these two changes, inference of the recurrent network can be performed in parallel over time for subsequent layer, reducing reloading of parameters into memory. In order to overcome the reduced performance from the change from more complex recurrent units to SRU's, 1-d convolutional layers are added between the recurrent units. The resulting system yields competitive performance with significantly reduced inference cost. This paper presents an interesting approach to ASR in a memory constrained setting and is well written. The results are strong, however additional runtime comparisons with architectures used for performance comparison would strengthen the empirical evidence supporting the utility of the method.

Reviewer 3



Summary: This paper proposes an acoustic model by combining simple recurrent units (SRUs) and simple 1-D convolution layers for multi-time step parallelization, which significantly improves the speed by overcoming the step-by-step constraint of RNNs. The proposed AM performs substantially better than SRUs alone and even better than LSTMs with the same number of parameters. It is combined with an RNN LM as a fully neural network-based speech recognizer. Word pieces are investigated as the model output targets in addition to graphemes, for further speedup in decoding. Quality: Strengths: - The idea of SRU + 1-D conv is simple but the improvement is impressive. - The multi-time step parallel processing for AM is shown to achieve nice speedup on a single core (6x with 8 parallel steps). - RNN LM with multi-stream processing is shown to be efficient enough to be used in beam search decoding, where it is typically used as second-pass rescoring due to efficiency constraint. - The paper shows a final model with total size 30MB is able to run in real time with a single core ARM. Weaknesses: - Do all the baseline LSTM models in the paper use recurrent projection units (Sak et al. 2014, Long Short-Term Memory Recurrent Neural Network Architectures for Large Scale Acoustic Modeling)? It should be easy to implement and it is effective to improve performance with the same number of parameters. - LSTM + 1-D conv should be used as the baseline in Table 3 and 5 since it is shown to be better than LSTM alone in Table 1. - It is kind of surprising that SRU + 1-D conv achieves better accuracy than LSTM + 1-D conv. It is worth discussing the reason / hypothesis in the paper. - In Table 6, it would be useful to also measure the computation time of LSTM AM for comparison. - The proposed system should be compared to a standard LSTM AM + WFST decoder graph with a (pruned) n-gram LM, which is shown to be efficient for embedded speech recognition in previous works (e.g. McGraw et al. 2016, Personalized speech recognition on mobile devices), especially now that the RNN LM is the computation bottleneck of the entire system as is shown in Figure 2. In fact, the speed of RNN LM could be further improved by 8-bit quantization, which typically does not degrade the WER. Clarity: In general, the presentation in the paper is clear and well-organized, except for the following questions: - In Table 2, what is the system performance with HCLM? - Does the system in Table 6 use characters or word pieces? Both should be shown for comparison. - How much is the WER different between different beam width in Figure 2? Does using half-precision in LM hurt WER? - It would be clearer to mention in Table 2 that 3.60 (for Deep Speech 2) is WER (instead of CER). Originality: The problem is well-motivated: multi-time step parallel processing for AM would be very useful for embedded system recognition, but SRU alone does not work. It is novel to find that simply combining 1-D conv with SRU substantially improves the quality of SRU, leading to better performance than LSTM. Other tricks and analyses in the paper to improve speed and accuracy are also useful. Significance: The results are very encouraging in terms of both accuracy and speed. Both the idea and the experiments would be of interest to the field and motivate future work. It is a good submission. I recommend an accept, although the issues mentioned above should be addressed.